# Comparison of Walking Quality Variables between End-Stage Osteonecrosis of Femoral Head Patients and Healthy Subjects by a Footscan Plantar Pressure System

**DOI:** 10.3390/medicina59010059

**Published:** 2022-12-28

**Authors:** Zehua Wang, Xingjia Mao, Zijian Guo, Ruipeng Zhao, Tengda Feng, Chuan Xiang

**Affiliations:** 1Department of Orthopedics, The Second Hospital of Shanxi Medical University, Taiyuan 030001, China; 2Center for Stem Cell and Regenerative Medicine, Department of Basic Medicine Sciences, Department of Orthopaedics of Sir Run Run Shaw Hospital, Zhejiang University School of Medicine, Hangzhou 310058, China

**Keywords:** osteonecrosis of the femoral head, gait analysis, plantar pressure system, plantar pressure distribution, symmetry index

## Abstract

*Background and Objectives*: Osteonecrosis of the femoral head (ONFH) is a progressive disease with a complex etiology and unknown pathogenesis. Gait analysis can objectively assess the functional behavior of the foot, thus revealing essential aspects and influencing factors of gait abnormalities. The aim of this study was to evaluate the differences in spatiotemporal parameters, static and dynamic plantar pressure parameters, and symmetry indices between patients with ONFH and healthy subjects. *Materials and Methods*: The study population consisted of 31 ONFH patients and 31 healthy volunteers. Gait parameters were obtained from the plantar pressure analysis system for both the ONFH and healthy groups. The symmetry index was calculated according to a formula, including spatiotemporal parameters, static and dynamic plantar pressure distribution, percentage of regional impulse, and percentage of the restricted contact area. *Results*: Compared with healthy controls, patients with ONFH had slower walking speed, shorter step length and stride length, and increased stride time, stance time, and percentage of stance. patients with ONFH had lower plantar static pressure on the affected side and higher contralateral plantar static pressure during stance than controls. During walking, the peak pressures in all regions on the affected side and the peak pressure in the toe 1 and metatarsal 3 regions on the healthy side were lower in ONFH patients than in controls. The percentage of contact area and regional impulse in the heel of both limbs were higher in ONFH patients than in the control group. The symmetry indexes of stride time, stance time, step length, maximum force, impulse and contacted area were significantly increased in ONFH patients compared to controls, with decreased symmetry. *Conclusions*: Osteonecrosis of the femoral head leads to characteristic changes in plantar pressure distribution. These changes may be interpreted as an attempt by patients with ONFH to reduce the load on the affected limb. Plantar pressure analysis may assist in the diagnosis of ONFH and can provide an objective quantitative indicator for the assessment of subsequent treatment outcomes.

## 1. Introduction

Osteonecrosis of the femoral head (ONFH) is a progressive disease of complex etiology and unknown pathogenesis, characterized by the disruption of blood supply, subchondral bone necrosis, and eventual femoral head collapse, resulting in severe hip pain and walking dysfunction, primarily in younger patients [1,2,3]. Patients with end-stage ONFH often undergo total hip arthroplasty (THA) to relieve pain and restore walking quality [4]. Although THA has become the most effective treatment for patients with end-stage ONFH, the results of THA are often not optimal in young adults or in the active population who undergo one or more revisions due to prosthesis loosening, excessive wear of the polyethylene prosthesis, and periprosthetic infections [5,6,7]. Early detection and timely intervention in ONFH are essential to slow its progression and improve the quality of life of patients.

The identification of minor differences that distinguish between healthy and abnormal patterns and highlight specific treatment responses are made feasible by tools that provide reliable and repeatable measurements in the diagnosis and developmental monitoring of diverse illnesses [8,9]. Gait analysis is an examination method that uses physical means to study walking patterns, which can objectively assess the functional behavior of the foot and thus reveal the important aspects of gait abnormalities and influencing factors [10]; therefore, it is considered a useful complement to clinical and imaging assessments [11,12,13]. In addition, gait analysis is also recommended in the Chinese guidelines for clinical diagnosis and treatment of osteonecrosis of the femoral head to objectively assess the effectiveness of the treatment of ONFH [14]. The plantar pressure system, as part of the gait analysis system, focuses on the performance of the foot in response to ground forces during daily activities [15]. The assessment of plantar pressure reveals how the first point of the kinematic chain connected to the leg is in contact with the ground and how the plantar region receives forces from the ground [16,17]. At the same time, it is the basis for analysis and measurement of abnormal plantar pressure distribution and gait, which is important for etiological analysis, diagnosis, functional and therapeutic evaluation of walking disorder-related diseases. In addition, measurements of static and dynamic plantar pressure distribution can highlight the characteristics of plantar pressure distribution in specific populations and help identify potential causes and the development of pathological gait through the comparative analysis with normal gait [18,19]. There have been many studies related to foot pressure in people with flat feet, vena cava foot, diabetic foot, stroke, obesity, osteoarthritis of the knee, and spinal cord injury [20,21,22,23,24,25,26]. There are no comparative studies on spatiotemporal parameters, plantar pressure distribution, and symmetry between patients with femoral head necrosis and normal subjects. Our aim was to describe and display these changes using the plantar pressure system so that we could determine whether the plantar pressure system could be a useful tool to display weight bearing and foot pressure problems in patients with ONFH, and at the end of our study, we attempted to obtain specific plantar pressure results in patients with ONFH that had not been previously studied.

Good walking quality is an indicator used to respond to an individual’s ability to walk, and it requires not only fast walking, but also symmetrical walking [27]. Walking symmetry is considered an important indicator for assessing walking quality in studies of stroke and unilateral limb injury [28,29]. In addition, there are numerous studies pointing to an increased risk of knee OA in patients with unilateral hip OA, especially in the contralateral knee [30,31,32]. However, it has not been determined whether there is an asymmetry in walking variables in patients with ONFH and the impact on other joints, which may be able to provide a sensitive indicator for the subsequent treatment and rehabilitation of patients with ONFH, thus improving their quality of life.

The aim of this study was to evaluate the differences in spatiotemporal parameters, static and dynamic plantar pressure parameters, and symmetry indices between patients with ONFH and healthy subjects, findings that may provide sensitive and quantitative parameters for diagnosis of the disease and subsequent therapeutic rehabilitation. Repeated plantar pressure measurements can be another way to reflect the severity of the disease and assess the effectiveness of treatment in patients with femoral head necrosis.

## 2. Materials and Methods

### 2.1. Study Design and Patients

The study was approved by the Ethics Committee of the Second Hospital of Shanxi Medical University (approval No. 2022YXNO. 169). Patients with ONFH who visited the Second Hospital of Shanxi Medical University from January 2021 to May 2022 were selected as the experimental group. Condition-matched healthy subjects were recruited from the community as the control group. The number of subjects included in each group was calculated to be 31 under the guidance of a statistical expert. All subjects signed an informed consent form before the start of the study. The inclusion criteria for the ONFH group were: (1) end-stage unilateral ONFH diagnosed according to the guidelines for clinical diagnosis and treatment of osteonecrosis of the femoral head in adults aged 18-80 years and confirmed as Grade III or IV according to the Association Research Circulation Osseous (ARCO); (2) The ability to walk independently for at least 15 min [14]. The exclusion criteria for the ONFH group were: (1) ONFH due to trauma; (2) History of lower extremity surgery; (3) Other surgery, cardiovascular disease, neuromuscular disease, or trauma that may affect gait; (4) Body index greater than 35; (5) Inability to walk independently; and (6) The need for assistive equipment such as crutches to assist walking [33]. The inclusion criteria for the control group were as follows: (1) no abnormalities in hip imaging; (2) no complaints of pain and discomfort in both lower extremities; and (3) baseline data should be matched with the experimental group. The exclusion criteria for the control group were the same as those for the experimental group.

In order to accurately evaluate the clinical diagnosis of the subjects, three experienced orthopedic clinicians assessed the grading of the condition by observing the subjects’ hip X-rays and MRIs.

### 2.2. Acquirement of Walking Pattern Data

Gait spatiotemporal plantar pressure parameters were collected using a plantar scanning pressure system (RSscan International, Olen, Belgium, 2096 mm × 472 mm × 18 mm, with 16384 resistive sensors arranged in a 256 × 64 matrix at a resolution of 2 sensors/cm^2^, data acquisition frequency: 125 Hz, pressure range: 0–200 N/cm^2^), which was connected to a computer with a supplied cable. The platform was located on a secure flat surface, leveled, and centered on a 10 m long rubber walkway. To prevent the subject from being frightened while walking on the scanning plate, a very thin, non-elastic cloth was placed over the surface of the plate. The examination room was evenly and softly lit to avoid excessive light that might affect the subject’s test results. The test system was calibrated before each measurement according to the manufacturer’s instructions [34].

Participants were informed of the purpose of the examination and precautions to be taken prior to the test, and they were asked to wear loose clothing that did not interfere with lower limb movement. The participants’ height and body mass were accurately measured prior to the gait test. The Footscan test system was then activated and basic information about the participants was entered, including name, gender, age, height, and body mass. Static plantar pressure parameters were first collected by having the participant stand in a natural state on the scanning platform. Each participant was then asked to perform an acclimatization walk along the track 5–10 times before the dynamic data was collected, always looking straight ahead during the walk to eliminate tension and to ensure that they passed through the test area with a natural and realistic gait. An appropriate initial position was determined for each participant based on stride and step length in the adaptation experiment to ensure that three consecutive walking cycles were completed and at least four steps were taken on the scanning board before passing through the test area. This minimizes the effect of acceleration and deceleration at the beginning and end of each walk on the examination results. Three complete gait assessments were performed on each participant, and the plantar pressure data were averaged from the three recordings.

### 2.3. Data Analysis

The plantar scanning plate system divides the sole of the foot into ten anatomical areas: (Ⅰ) toe 1 (T1), (Ⅱ) toes 2 to 5 (T2-5), (Ⅲ) metatarsal 1 (M1), (Ⅳ) metatarsal 2 (M2), (Ⅴ) metatarsal 3 (M3), (Ⅵ) metatarsal 4 (M4), (Ⅶ) metatarsal 5 (M5), (Ⅷ) midfoot (MF), (Ⅸ) heel medial (HM), and (Ⅹ) heel lateral (HL) (Figure 1) [26].

Data on gait spatiotemporal parameters (walking speed, stride time, stance time, stride length and step length), static plantar pressure (forefoot, hindfoot and total foot), and dynamic plantar pressure (maximum force (Max F), impulse and contact area of each region) were collected by Footscan 7 Gait software. The side with radiographic features of osteonecrosis of the femoral head was defined as the affected side, and the other was defined as the healthy side. The mean of the above gait parameters was calculated.

In order to eliminate individual differences and make the gait parameters comparable, a subset of gait parameters was normalized as a percentage of the part parameter value to the sum of the parameter values.

The percentage of stance time to stride time was calculated as stance phase percentage:(1)$$Stance\;phase\;percentage\left(\%\right)=100\%\times \frac{Stance\;time\left(s\right)}{\mathrm{Stride}\;\mathrm{time}\left(s\right)}$$

The percentage of the regional impulse of the 10-distribution areas to the total of the entire area was calculated as the regional impulse percentage:(2)$$Regional\;percentage\left(\%\right)=100\%\times \frac{\mathrm{Impulse}\;\mathrm{value}\;\mathrm{of}\;\mathrm{each}\;\mathrm{region}}{\mathrm{Impulse}\;\mathrm{value}\;\mathrm{of}\;\mathrm{the}\;\mathrm{whole}\;\mathrm{foot}}$$

The percentage of the regional contact area of the 10-distribution areas to the total of the entire foot contact area was calculated as the regional contact area percentage:(3)$$Regional\;contact\;percentage\left(\%\right)=100\%\times \frac{\mathrm{Contact}\;\mathrm{area}\;\mathrm{of}\;\mathrm{each}\;\mathrm{region}}{\mathrm{Contact}\;\mathrm{area}\;\mathrm{of}\;\mathrm{the}\;\mathrm{whole}\;\mathrm{foot}}$$

The following formula was used to calculate the gait parameter’s symmetry index (SI). Stance time SI, step length SI, Max F SI, impulse SI, and contact area SI were calculated separately.
(4)$$Symmetry\;Index=\left|1-\frac{Variables\;in\;step\;length\;short\;side}{Variables\;in\;step\;length\;longer\;side}\right|\;$$

### 2.4. Statistical Analysis

The data were expressed as the mean ± standard deviation. Statistical analyses used SPSS software version 23.0 (IBM, Armonk, NY, USA). The variables were investigated using the Kolmogorov-Smirnov test to determine whether they were normally distributed. Comparisons of experimental outcomes between ONFH and healthy groups were undertaken using two independent samples. The tests used were the *t*-test (normally distributed parameters) or the Mann-Whitney U-test (non-normally distributed parameters). A two-tailed paired t-test was applied to assess the variation between the left and right sides. Values with *p* < 0.05 were considered statistically significant.

## 3. Results

### 3.1. Baseline Demographics

Data from 62 participants, comprising 31 healthy and 31 ONFH patients, were evaluated. The study discovered no statistically significant variations in baseline demographics between the healthy and ONFH groups (*p* > 0.05; Table 1). There were no significant variations in the gait parameter of the bilateral limbs in the control group (*p* > 0.05; See in Appendix A Appendix A).

### 3.2. Spatiotemporal Variables in ONFH and Healthy Groups

The walking speed in the ONFH group (0.61 ± 0.20m/s) was significantly slower than in the healthy group (1.11 ± 0.24m/s) (*p* < 0.05; Table 2). The stride time and stance time were significantly longer in the ONFH group than those in the healthy group (*p* < 0.05; Table 2). The stance phase percentage was also significantly larger in the ONFH group than in the healthy one (72.35 ± 6.21%). The stride length and step length were both shorter in the ONFH group than for those in the healthy group (*p* < 0.05; Table 2).

### 3.3. Plantar Pressure Distribution in the ONFH and Healthy Groups

#### 3.3.1. Static Plantar Pressure Distribution in ONFH and Healthy Group

The afflicted side was defined as the side having imaging characteristics of femoral head necrosis and the other side as the healthy side. The parameters of the affected side and the healthy side of the ONFH group were compared with those in the healthy group separately.

Compared with the healthy group, the total plantar static pressure and hindfoot plantar pressure of the affected limb in the ONFH group were significantly lower than those of the healthy group, while the total plantar static pressure and hindfoot plantar pressure of the healthy limb were significantly higher than those of the healthy group (*p* < 0.05; Table 3).

#### 3.3.2. Dynamic Plantar Pressure Distribution in ONFH and Healthy Group

The highest peak plantar pressure in the healthy group and on both sides of ONFH patients was located in the third metatarsal region (M3). In T1 and M3 of the healthy side, the peak plantar pressure was significantly decreased in the ONFH group compared with the healthy group. The peak plantar pressure in each region on the affected side was lower in the ONFH group than in the healthy group, and the differences were significant (*p* < 0.05; Table 4).

### 3.4. Regional Impulse Percentage and Contact Area Percentage in Different Regions of ONFH and Healthy Group

Significantly increased impulse percentage under the heel (HM and HL) was found in ONFH patients. Under bilateral M3, M5, and M2 of the affected side, the increased impulse was significantly decreased (*p* < 0.05; Table 5).

In M1, HM, and HL on both sides and M2 of the healthy side, the regional contact area percentage in ONFH patients was significantly increased compared with that in healthy subjects. Bilateral M5 regional contact area percentage in ONFH patients was significantly decreased compared with that in healthy subjects (*p* < 0.05; Table 6).

### 3.5. Symmetry Index of ONFH and Healthy Groups

The symmetry index (SI) of the stride time, stance time, and step length were significantly higher in the ONFH group compared to the healthy group (*p* < 0.05; Table 7). In addition, the SI of the whole foot plantar Max F, impulse, and contact area were significantly greater in the ONFH group than in the healthy group (*p* < 0.05; Table 7).

## 4. Discussion

Plantar pressure measurements can be used to objectively assess the functional behavior of the foot and help to resolve kinematic and kinetic biases [11,12,13,25] and are considered a useful complement to clinical and imaging assessments [35]. The purpose of this study was to elucidate if there were any differences in spatiotemporal parameters and plantar pressure parameters between patients with ONFH and healthy subjects. Thus, we performed several measurements of spatiotemporal and plantar pressure parameters in both groups of subjects using the footscan plantar pressure system, and the results of the study revealed that, compared with healthy controls, patients with ONFH had slower walking speed, shorter step length and stride length, and increased stride time, stance time, and percentage of stance. Patients with ONFH had lower plantar static pressure on the affected side and higher contralateral plantar static pressure during stance than controls. During walking, the peak pressures in all regions on the affected side and the peak pressure in the toe 1 and metatarsal 3 regions on the healthy side were lower in ONFH patients than in controls. The percentage of contact area and regional impulse in the heel of both limbs were higher in ONFH patients than in the control group. The symmetry indexes of stride time, stance time, step length, maximum force, impulse, and contacted area were significantly increased in ONFH patients than in controls, with decreased symmetry.

Adequate walking speed is a key factor in maintaining the body’s exercise routine [36,37]. Ismailidis et al. used the inertial sensor system RehaGait to collect gait data from 22 patients with hip osteoarthritis, and found that subjects with hip osteoarthritis walked at a slower speed and had a significantly shorter percentage of stride length and single support time on the affected side than the healthy group [38]. A study by Porta et al. included 11 patients diagnosed with grade IV hip osteoarthritis and 11 healthy controls, which were analyzed kinematically using a motion capture system consisting of eight infrared cameras. The results of the study showed that the gait of patients with hip osteoarthritis was characterized by reduced walking speed, stride length, stride frequency, swing phase time, an increased support phase and double support time [39], a result that is parallel to the results in our study. In our study, walking speed was slower in the ONFH group than in the healthy group; stride time and stance time were significantly longer in patients in the ONFH group than in the healthy group, and the stance phase percentage was greater than in the healthy group, suggesting a shorter propulsive swing phase in the ONFH group. Patients with ONFH may suffer from hip pain, decreased mobility, and hip muscle contracture weakness, thereby causing rapid limb swing and reduced stride length.

There are no comparable studies of dynamic and static plantar pressure distribution in subjects with ONFH. Compared to healthy controls, the static plantar pressure in the affected limb was significantly lower in ONFH patients than in healthy subjects, and was mainly manifested by a decrease in hindfoot pressure. At the same time, the healthy limb was burdened with more trunk weight, as evidenced by increased static plantar pressure in the hindfoot. Previous research employing the plantar pressure system on plantar pressure patterns during normal walking in healthy people found that the highest average plantar pressure occurred below the third metatarsal head, and the second highest average plantar pressure occurred below the hindfoot [19,26,40]. This is consistent with the findings in healthy subjects in this study, where the plantar pressure distribution in ONFH patients had the same characteristics as healthy subjects, but the peak pressures were lower in all regions, suggesting a specific bout of pain management strategy to reduce the forces exerted on the affected joint in ONFH patients. In addition, in the heel region (HM and HL), the impulse percentage and the regional contact area percentage were higher in ONFH patients than in healthy controls, and a study in multiple sclerosis yielded similar results [41]; it showed that the decrease of plantar flexion and forward propulsion forces in the swing phase and the increase of the plantar contact area are inevitable due to the loss of motor control of the limb during heel landing. In the present study, we believe that the increase in the plantar contact area and the increase in impulse volume are still largely associated with reduced joint range of motion and pain.

Variables such as space, time, and plantar pressure were all assessed concurrently in this research. The current assessment of walking quality by walking speed alone is inadequate. Recently, symmetry variables have been indicated as a key indicator for assessing walking ability [42]. Healthy individuals have a symmetrical gait walking pattern through spatiotemporal parameters under normal motor control as well as muscle activation that is symmetrical [43,44]. Walking asymmetry, on the other hand, is regarded as a significant feature in measuring walking quality in patients with unilateral limb impairments caused by stroke and amputation [28,29,42,45,46,47,48]. We did not find any relevant studies on walking asymmetry in patients with ONFH, and the study by el-Gamal et al. showed the presence of walking asymmetry in patients with unilateral hip osteoarthritis compared to the healthy population [49], which is parallel to the results of our study. The formula used to calculate symmetry in our study was the same as the previous formula used for patients with stroke and spinal cord injury, where a smaller value of SI indicated better symmetry, and the results of the study showed that patients with unilateral femoral head osteonecrosis had not only spatial asymmetry (step length SI), but also temporal asymmetry (step time SI, stride time SI), as well as plantar pressure asymmetry (Max F SI, impulse SI) and plantar contact area asymmetry. When a subject is walking, the limb in the support phase provides support and control for the limb in the swing phase on the opposite side so that the limb in the swing phase on the other side is able to move forward. The asymmetry may be due to the collapsed deformity of the femoral head on the affected side, limb shortening, and hip dysfunction, all of which prevent the affected limb from bearing the weight of the entire torso in its supporting phase, resulting in dysfunction of the contralateral limb in the swing phase.

In this study, we used a plantar pressure system to collect gait spatiotemporal and plantar pressure distribution parameters from patients with ONFH. The data collection process was relatively simple and easy, and the guidance of professional researchers and instrument technicians made the gait data reliable and realistic. However, there are several limitations of this study. First, the small sample size made it difficult to stratify the analysis according to the degree of femoral head necrosis. Second, only patients with ARCO stages 3 and 4 were included in the ONFH group of this study because there are no characteristic clinical manifestations of early femoral head necrosis, and therefore most patients with early ONFH do not visit the hospital, making it difficult to include a sufficient number of patients with early ONFH in a short period of time. In addition, this study did not standardize the gender of the included subjects, and differences in gait parameters between genders have been suggested in previous studies. Therefore, in subsequent studies, the sample size should be increased according to the specific research questions, especially for patients with ARCO stage I and II, to stratify patients with different levels of ONFH as a way to further observe the differences in plantar pressure distribution in different sub-stages. In addition, an attempt can be made to study whether gender has an effect on the plantar pressure distribution of ONFH patients to some extent. These factors will be beneficial in furthering the understanding of the characteristics of plantar pressure distribution in patients with ONFH.

## 5. Conclusions

In conclusion, we used the plantar pressure system to identify differences in spatiotemporal parameters, plantar pressure distribution, and symmetry between patients with ONFH and healthy subjects. Osteonecrosis of the femoral head leads to characteristic changes in plantar pressure distribution. These changes may be interpreted as an attempt by patients with ONFH to reduce the load on the affected limb. Plantar pressure analysis may assist in the diagnosis of ONFH and can provide an objective quantitative indicator for the assessment of subsequent treatment outcomes.

## Figures and Tables

**Figure 1 medicina-59-00059-f001:**
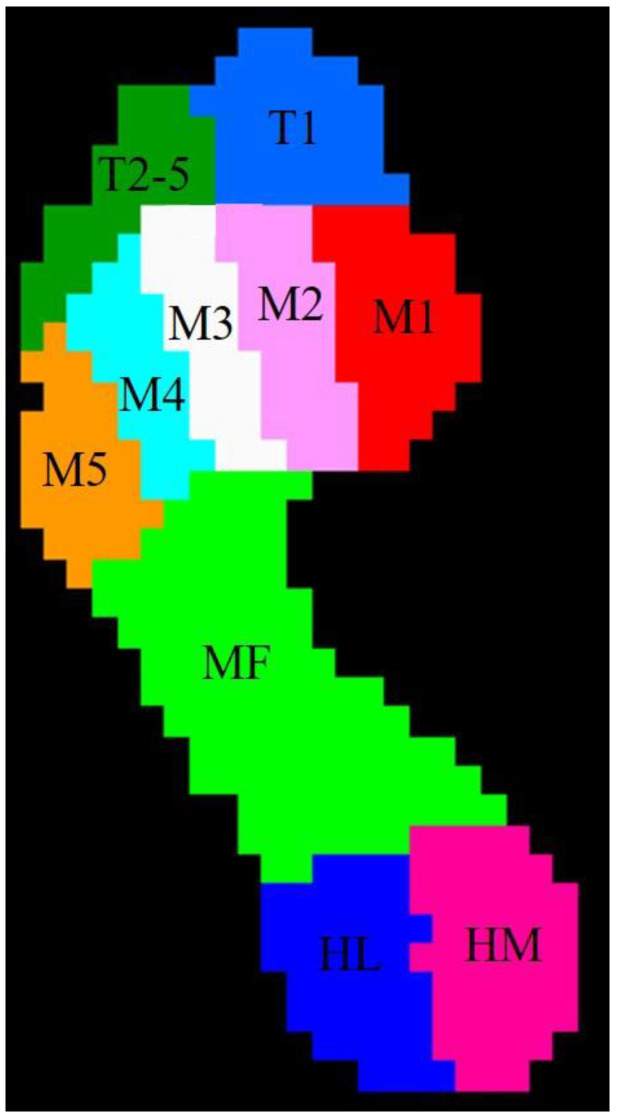
The plantar pressure system divides the foot into 10 regions: (Ⅰ) toe 1 (T1), (Ⅱ) toes 2 to 5 (T2-5), (Ⅲ) metatarsal 1 (M1), (Ⅳ) metatarsal 2 (M2), (Ⅴ) metatarsal 3 (M3), (Ⅵ) metatarsal 4 (M4), (Ⅶ) metatarsal 5 (M5), (Ⅷ) midfoot (MF), (Ⅸ) heel medial (HM), and (Ⅹ) heel lateral (HL).

**Table 1 medicina-59-00059-t001:** Demographics of subjects with osteonecrosis of the femoral head (ONFH) and healthy subjects.

Items	Healthy Group	ONFH Group	*p* Value
Age (mean ± SD, years)	54.71 ± 12.07 ^b^	56.00 ± 13.20 ^b^	0.69
Female/male (n)	14/17	17/14	1
Height (mean ± SD, cm)	163.77 ± 6.54 ^b^	164.77 ± 9.83 ^a^	0.64
Body mass (mean ± SD, kg)	66.19 ± 8.42 ^a^	70.06 ± 10.14 ^a^	0.11
Body mass index (mean ± SD, kg/m^2^)	24.66 ± 2.66 ^a^	25.85 ± 3.46 ^a^	0.14

There was no significant difference in the baseline demographics between the healthy and ONFH groups (*p* > 0.05). ^a^: normal distribution. ^b^: non-normal distribution.

**Table 2 medicina-59-00059-t002:** Spatiotemporal variables in the ONFH group and healthy group.

Variables	Healthy Group	ONFH Group	*p* Value
Walking speed (m/s)	1.11 ± 0.24 ^a^	0.61 ± 0.20 ^a,^*	<0.01
Stride time (s)	1.09 ± 0.10 ^a^	1.30 ± 0.16 ^b,^*	<0.01
Stance time (s)	0.73 ± 0.07 ^a^	0.93 ± 0.12 ^a,^*	<0.01
Stance phase percentage (%)	66.88 ± 5.29 ^a^	72.35 ± 6.21 ^a,^*	<0.01
Stride length (m)	1.05 ± 0.10 ^a^	0.76 ± 0.19 ^a,^*	<0.01
Step length (m)	0.53 ± 0.05 ^a^	0.37 ± 0.10 ^b,^*	<0.01

* *p* < 0.05, vs. healthy group (mean ± SD; independent-sample *t*-test). ONFH: osteonecrosis of the femoral head; m: meter; s: second. ^a^: normal distribution. ^b^: non-normal distribution.

**Table 3 medicina-59-00059-t003:** Static plantar pressure distribution in ONFH group and healthy group.

	Healthy Group	Affected Side of ONFH Group	Healthy Side of ONFH Group	*p* Value	
				*p ^a^*	*p ^b^*
Forefoot (%)	21.49 ± 2.82 ^a^	20.19 ± 3.65 ^a^	23.24 ± 6.89 ^a^	0.12	0.20
Hindfoot (%)	29.87 ± 3.64 ^a^	23.67 ± 5.67 ^a,^*	32.90 ± 5.88 ^a,^*	<0.01	0.02
Total (%)	51.36 ± 4.45 ^a^	43.87 ± 6.99 ^a,^*	56.13 ± 6.99 ^a,^*	<0.01	<0.01

* *p* < 0.05, vs. healthy group (mean ± SD; independent-sample *t*-test). ONFH: osteonecrosis of the femoral head. ^a^: normal distribution. ^b^: non-normal distribution. *p*
^a^: affected side of ONFH group vs. healthy group; *p*
^b^: healthy side of ONFH group vs. healthy group.

**Table 4 medicina-59-00059-t004:** Dynamic plantar pressure distribution in ONFH group and healthy group.

	Healthy Group	Affected Side of ONFH Group	Healthy Side of ONFH Group	*p* Value	
				*p ^a^*	*p ^b^*
T1(N/cm^2^)	6.33 ± 2.42 ^a^	4.85 ± 2.84 ^a,^*	4.75 ± 2.95 ^a,^*	0.30	0.02
T2-5(N/cm^2^)	2.37 ± 1.80 ^b^	1.47 ± 1.21 ^b,^*	2.03 ± 1.82 ^b^	0.02	0.47
M1(N/cm^2^)	7.12 ± 3.58 ^a^	5.36 ± 3.19 ^a,^*	7.36 ± 4.28 ^a^	0.04	0.81
M2(N/cm^2^)	13.28 ± 3.76 ^b^	10.15 ± 5.0 ^a,^*	11.61 ± 4.88 ^a^	<0.01	0.14
M3(N/cm^2^)	17.51 ± 5.18 ^a^	12.42 ± 4.89 ^a,^*	11.98 ± 5.33 ^a,^*	<0.01	<0.01
M4(N/cm^2^)	11.60 ± 4.4 ^a^	8.73 ± 4.05 ^a,^*	10.23 ± 5.44 ^b^	0.01	0.28
M5(N/cm^2^)	7.17 ± 4.69 ^b^	4.98 ± 3.59 ^b,^*	6.43 ± 4.15 ^b^	0.04	0.51
MF(N/cm^2^)	4.53 ± 1.73 ^b^	3.61 ± 1.4 ^a,^*	3.78 ± 1.26 ^a^	0.03	0.06
HM(N/cm^2^)	9.96 ± 2.28 ^a^	8.58 ± 2.72 ^a,^*	10.59 ± 3.27 ^a^	0.04	0.38
HL(N/cm^2^)	10.22 ± 3.22 ^b^	8.14 ± 2.41 ^a,^*	9.52 ± 3.07 ^a^	<0.01	0.38

The footscan plate system partitioned the foot into the following ten anatomical regions: (Ⅰ) toe 1 (T1), (Ⅱ) toes 2 to 5 (T2-5), (Ⅲ) metatarsal 1 (M1), (Ⅳ) metatarsal 2 (M2), (Ⅴ) metatarsal 3 (M3), (Ⅵ) metatarsal 4 (M4), (Ⅶ) metatarsal 5 (M5), (Ⅷ) midfoot (MF), (Ⅸ) heel medial (HM), and (Ⅹ) heel lateral (HL). * *p* < 0.05, vs. healthy group (mean ± SD; independent-sample t-test). ONFH: osteonecrosis of the femoral head. ^a^: normal distribution. ^b^: non-normal distribution. *p*
^a^: affected side of ONFH group vs. healthy group; *p*
^b^: healthy side of ONFH group vs. healthy group.

**Table 5 medicina-59-00059-t005:** Regional impulse percentage (%) in different regions of ONFH and healthy groups.

	Healthy Group	Affected Side of ONFH Group	Healthy Side of ONFH Group	*p* Value	
				*p ^a^*	*p ^b^*
T1(%)	6.20 ± 4.46 ^b^	6.52 ± 5.66 ^b^	5.30 ± 4.11 ^b^	0.80	0.42
T2-5(%)	1.46 ± 1.43 ^b^	1.36 ± 1.66 ^b^	1.60 ± 2.33 ^b^	0.79	0.78
M1(%)	6.63 ± 3.71 ^a^	8.48 ± 6.37 ^b^	8.02 ± 4.40 ^a^	0.17	0.18
M2(%)	12.33 ± 3.05 ^a^	10.05 ± 4.41 ^a,^*	11.69 ± 3.83 ^a^	0.02	0.48
M3(%)	13.01 ± 3.64 ^a^	9.93 ± 3.48 ^a,^*	10.06 ± 4.03 ^a,^*	<0.01	<0.01
M4(%)	9.29 ± 3.29 ^a^	7.73 ± 3.78 ^a^	8.65 ± 3.66 ^a^	0.09	0.47
M5(%)	7.38 ± 3.91 ^b^	5.20 ± 3.82 ^b,^*	5.49 ± 3.40 ^b,^*	0.03	0.05
MF(%)	18.68 ± 7.72 ^a^	17.84 ± 10.01 ^b^	16.39 ± 6.18 ^b^	0.72	0.20
HM(%)	13.50 ± 4.13 ^a^	19.22 ± 7.09 ^a,^*	18.86 ± 7.63 ^b,^*	<0.01	<0.01
HL(%)	11.43 ± 3.53 ^a^	13.54 ± 5.53 ^b,^*	13.78 ± 5.01 ^b,^*	0.08	0.04

The footscan plate system partitioned the foot into the following ten anatomical regions: (Ⅰ) toe 1 (T1), (Ⅱ) toes 2 to 5 (T2-5), (Ⅲ) metatarsal 1 (M1), (Ⅳ) metatarsal 2 (M2), (Ⅴ) metatarsal 3 (M3), (Ⅵ) metatarsal 4 (M4), (Ⅶ) metatarsal 5 (M5), (Ⅷ) midfoot (MF), (Ⅸ) heel medial (HM), and (Ⅹ) heel lateral (HL). * *p* < 0.05, vs. healthy group (mean ± SD; independent-sample t-test). ONFH: osteonecrosis of the femoral head. ^a^: normal distribution. ^b^: non-normal distribution. *p*
^a^: affected side of ONFH group vs. healthy group; *p*
^b^: healthy side of ONFH group vs. healthy group.

**Table 6 medicina-59-00059-t006:** Regional contact area percentage (%) in different regions of ONFH and healthy groups.

	Healthy Group	Affected Side of ONFH Group	Healthy Side of ONFH Group	*p* Value	
				*p ^a^*	*p ^b^*
T1(%)	10.53 ± 1.76 ^a^	9.60 ± 2.32 ^b^	9.25 ± 3.56 ^a^	0.08	0.08
T2-5(%)	7.24 ± 3.29 ^b^	6.71 ± 3.55 ^a^	6.81 ± 3.53 ^a^	0.55	0.62
M1(%)	7.54 ± 1.22 ^b^	9.29 ± 2.45 ^a,^*	9.21 ± 2.11 ^a,^*	<0.01	<0.01
M2(%)	6.93 ± 0.51 ^b^	7.12 ± 1.81 ^b^	7.40 ± 0.88 ^a,^*	0.58	<0.01
M3(%)	5.65 ± 0.44 ^b^	5.58 ± 1.40 ^b^	5.86 ± 0.88 ^a^	0.80	0.23
M4(%)	5.69 ± 0.51 ^b^	5.57 ± 1.49 ^b^	5.76 ± 1.13 ^a^	0.69	0.73
M5(%)	7.54 ± 1.13 ^b^	6.34 ± 1.95 ^b,^*	6.39 ± 1.72 ^a,^*	<0.01	<0.01
MF(%)	28.12 ± 2.87 ^b^	27.23 ± 7.01 ^b^	26.99 ± 4.67 ^b^	0.52	0.26
HM(%)	11.26 ± 1.27 ^b^	12.18 ± 1.83 ^a,^*	12.10 ± 1.26 ^a,^*	0.03	<0.01
HL(%)	9.50 ± 1.14 ^b^	10.37 ± 1.69 ^a,^*	10.23 ± 1.17 ^a,^*	0.02	<0.01

The footscan plate system partitioned the foot into the following ten anatomical regions: (Ⅰ) toe 1 (T1), (Ⅱ) toes 2 to 5 (T2-5), (Ⅲ) metatarsal 1 (M1), (Ⅳ) metatarsal 2 (M2), (Ⅴ) metatarsal 3 (M3), (Ⅵ) metatarsal 4 (M4), (Ⅶ) metatarsal 5 (M5), (Ⅷ) midfoot (MF), (Ⅸ) heel medial (HM), and (Ⅹ) heel lateral (HL). * *p* < 0.05, vs. healthy group (mean ± SD; independent-sample t-test). ONFH: osteonecrosis of the femoral head. ^a^: normal distribution. ^b^: non-normal distribution. *p*
^a^: affected side of ONFH group vs. healthy group; *p*
^b^: healthy side of ONFH group vs. healthy group.

**Table 7 medicina-59-00059-t007:** Comparison of ONFH and healthy groups.

Variables	Healthy Group	ONFH Group	*p* Value
Stride time SI	0.02 ± 0.01 ^b^	0.05 ± 0.04 ^b,^*	<0.01
Stance time SI	0.04 ± 0.05 ^b^	0.10 ± 0.09 ^b,^*	<0.01
Step length SI	0.04 ± 0.04 ^a^	0.15 ± 0.16 ^b,^*	<0.01
Max F SI	0.09 ± 0.06 ^b^	0.19 ± 0.17 ^b,^*	<0.01
Impulse SI	0.11 ± 0.06 ^a^	0.22 ± 0.20 ^b,^*	<0.01
Contact area SI	0.04 ± 0.03 ^a^	0.08 ± 0.10 ^b,^*	0.04

* *p* < 0.05, vs. healthy group (mean ± SD; independent-sample *t*-test). SI: Symmetry index; ONFH: osteonecrosis of the femoral head. ^a^: normal distribution. ^b^: non-normal distribution.

## Data Availability

Data available on request from the authors. The data that support the findings of this study are available from the corresponding author, [C.X.], upon reasonable request.

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
