# Peer review of "Comparison of Walking Quality Variables between End-Stage Osteonecrosis of Femoral Head Patients and Healthy Subjects by a Footscan Plantar Pressure System"

_medicina, 2022, doi:10.3390/medicina59010059_

Round 1

Reviewer 1 Report

Thank you for submitting this manuscript. While this is an interesting body of work that has the potential to have clinical implications, much of the manuscript needs to be clarified and further information included to improve its quality prior to publication. The grammar of the manuscript should also be improved prior to publication.

Please see my detailed comments and suggestions below.

Abstract

Page 1, Lines 23-24: When you say that ‘plantar pressures decreased on the affected side and increased on the contralateral side’ do you mean with time, in comparison to controls, or that plantar pressures were lower on the affected side than the contralateral side during stance? This is unclear.

Page 1, Lines 24-27: Similarly, in the following sentence, did peak pressures decrease over time during the gait cycle, or in comparison to controls? You should also specify whether the reference to percentage of peak impulse is with regards to the control group.  

Page 1, Lines 27-28: Please explain what you mean by the SI being greater in both ONFH patients. Were there 2 separate groups/only 2 patients? This needs to be clarified.

Page 1, Lines 28-31: Is the conclusion based on the ability to discern differences between the groups in your study? If so, please clarify this in this section.

Introduction

Page 2, Line 42: Remove the word ‘however’ here as it is redundant.

Page 2, Lines 47-49: Please re-word this sentence as it is difficult to follow because of the grammar.

Page 2, Line 54: Recommended by who?

Page 2, Line 74: Please explain what you mean by an ‘excellent walking quality’.

General comment: The Introduction contains many long sentences that make it difficult to follow in places. To improve this section, I suggest making this section more succinct, and checking the grammar with a professional translator.

Methods

General comment: Please explain how the sample size was calculated. This is important to be able to interpret the results.

Page 3, Lines 92-94: Where were the control group recruited from? The methods suggest from the hospital, but how were they identified from a hospital if they were healthy controls?

Page 3, Lines 96-100: Was the participant eligibility confirmed by a clinician or a researcher? If confirmed by a researcher, were they appropriately trained to be able to know if someone fitted the first criteria for the study (i.e. their clinical diagnosis)?

Page 3, Line 103: Why were patients with BMI >35 excluded?

Page 3, Line 109-110: Please correct this sentence as it is difficult to follow. You should also explain who assessed the eligibility of healthy volunteers. Were these clinicians or researchers? It appears some clinical knowledge was required to adequately assess eligibility.

Page 3, Line 112: I suggest using different terminology to ‘walking pattern’ here ,as it suggest you were collecting kinematic data, which is not the case.

Page 5, Line 153: Please write SD in full here, as it is the first use of the abbreviation.

General comment: Please check the grammar of this section as some sentences are difficult to follow at present due to the poor grammar.

Results:

Page 5, Lines 165-166: Where is the data to support your statement that there are no statistical differences between the control group’s limbs?

General comment: How many steps on average did participants take when walking across the walkway?

Discussion:

Page 10, Line 332: What makes the data reliable and realistic? Please elaborate.

General comment: Please avoid repeating the results section in the discussion, and simply discuss your findings with respect to previous studies. You should also ensure that you cite previous research appropriately. Much of the discussion on page 10 has no references to support the content.

Page 10, Line 325: Please define ARCO. This is the first use of the abbreviation.

Page 10, Lines 331-332: It is not correct that ‘as many patients as possible’ should be recruited, as it is ethically not appropriate. The study should have enough patients to answer the research question. This is one of the reasons a sample size calculation is important.

General comment:  The implications of the findings need to be better addressed in the Discussion and Conclusion.

Tables and Figures:

For all tables, please present the p-values for each of the variables you compared statistically in each of the tables. It is not good enough to present >0.05 or <0.05 in the manuscript. The true p-values need to be presented for all analyses described. You should also ensure that all tables have the variable’s units described. For example, these are missing in Table 7.

Reviewer 2 Report

Reviewer’s comments for the re-submitted article titled “Comparison of walking quality variables between end-stage osteonecrosis of femoral head patients and healthy subjects by a footscan plantar pressure system”

 Dear Editor

 Thank you for giving me the opportunity to review the revised version of the manuscript “Comparison of walking quality variables between end-stage osteonecrosis of femoral head patients and healthy subjects by a footscan plantar pressure system”. The authors of this study investigated the differences between patients with osteonecrosis over healthy individuals based on spatial and dynamic (plantar pressure distribution) parameters as well as the symmetry of gait. The study was sufficiently designed, and the results were quite informative and adequately discussed. Some points could, however, be clarified so that the reader was able to understand and be able to reproduce the experimental conditions, under which the study was conducted.

Lines 56-61: Please consider spiting this sentence in two parts (preferably in line 58 after “… from the ground [16,17].”)

Line 83: The authors refer throughout the paper to static and dynamic data of plantar pressure distribution. As it is not clear, they should consider clarifying the differences between the static and dynamic parameters of plantar pressure distribution preferably in the Materials and Methods section.

Line 94: Please replace “…weight…” with “…body mass…”.

Lines 97-100: The authors should consider clarifying the “…clear imaging evidence…” based on which ONFH patients were included in this study. This is important to be noted in this section, as the reader becomes aware of the severity of osteonecrosis of the included patients only at the end of the manuscript, where the authors refer to this as a limitation of the study. It appears that only patients with ARCO 3 and 4 stages were included in this study.

Lines 109-110: Please delete the sentence “Subjects were consulted and evaluated demographic…” as it is repeated immediately after.

Lines 112-113: Please provide more information on the foot-scanning pressure plate system (e.g., sampling frequency, way of data transfer, supporting equipment etc.). It would be quite informative if the authors could provide a picture of the experimental setup.

Line 119: Please define “…proper acclimatization…” How many steps or how much time participants had to practice to achieve this?

Line 120: How many steps were eventually performed by each participant in each trial (considering the length of the foot-scanning pressure plate system) based on which the average was calculated?

Lines 130-135: The authors should consider placing all parameters in a Table with a brief description of what each of these parameters represented as some readers may not be familiar with some of them (e.g., impulse, contact area walking speed etc.). Additionally, static, and dynamic parameters should be more self-explanatory (see comment above) as there is no mention of how they are distinguished from each other.

Line 155: Please replace “…usually…” with “…normally…”

Line 157: Please replace “…customarily…” with “…normally…”

Line 162: Since the authors investigated whether data were normally distributed or not, please provide the relevant information in the Results section.

Line 167: Please replace “Weight” with “Body mass” in Table 1

Line 173: Please add units after … (1.11±0.24) …

Line 176: Please add units after … (72.35±6.21) …

Line 173-174: The authors should consider replacing “The stride time was significantly longer, and the stance time was significantly larger…” with “The stride and stance time was significantly longer…” as the adjective “larger” is usually used for length.

Lines 251-256: Please consider spiting this sentence into two parts.

Lines 258, 261, 302: Unless it is required by the journal, please delete the initials from the authors’ names in the text.

Line 309: Please replace “…and plantar pressure…” with “…as well as plantar pressure…”

Lines 312-318: This is a quite large sentence. Please consider splitting it into two or three sentences.

Line 323: Do the authors have information to show that gait data was actually reliable?

Line 325: Please explain ARCO

Line 346: Please explain “…meta region…”

Reviewer 3 Report

Suggestions are attached 

Round 2

Reviewer 3 Report

In the last two decades guidleiness invaded almost all the medical disciplines. The outcome is total devastation to the art of medicine, in other words its a mortal strategy and the worst invention applied to medicine. 

In my opinion, the article has nothing to do with genuine clinical practice, its  a technical work and never touches the core of the scientific clinical potential.